# Post-Reproductive Lifespan in Honey Bee Workers with Varying Life Expectancies

**DOI:** 10.3390/ani15101402

**Published:** 2025-05-13

**Authors:** Karolina Kuszewska

**Affiliations:** Department of Zoology and Animal Welfare, Faculty of Animal Science, University of Agriculture in Krakow, Al. Mickiewicza 21, 31-120 Krakow, Poland; karolina.kuszewska@urk.edu.pl

**Keywords:** *Apis mellifera*, division of labor, menopause, post-reproductive lifespan, social insect, grandmother hypothesis, ovaria

## Abstract

This study investigates the post-reproductive lifespan and ovarian activation in honeybee workers (*Apis mellifera*) with different life expectancies, based on the “grandmother hypothesis”. We divided newly emerged bees into a control group and an injured group with shortened lifespans due to thorax puncturing. Observations of ovarian development and hypopharyngeal gland size were recorded. The results showed that injured bees had significantly shorter lifespans, and gland size varied with age and treatment, indicating injuries affected development. Notably, older bees exhibited reduced ovarian activation, suggesting a form of menopause, where energy shifts from reproduction to colony contribution as they age.

## 1. Introduction

The evolution of the post-reproductive lifespan in humans and other animals is a fascinating topic that has garnered significant attention in the field of evolutionary biology [1,2,3]. One of the primary explanations for this phenomenon revolves around the indirect fitness advantages that older females can gain by investing their time and resources in the care of their offspring and even their grandchildren [1]. This concept, often referred to as the “grandmother hypothesis”, suggests that while older females may no longer reproduce, their presence and assistance can greatly enhance the survival and reproductive success of their descendants. Evidence supporting the adaptive significance of the post-reproductive lifespan in humans can be traced back to demographic studies conducted in Finland and Canada during the 18th and 19th centuries [4]. These studies revealed a strong correlation between the longevity of grandmothers and the reproductive success of their children. Specifically, it was found that grandmothers who lived longer tended to have grandchildren who thrived and reproduced more successfully [4]. This suggests that the role of grandmothers is not merely passive but actively contributes to the survival of the next generation, thereby enhancing the overall fitness of the family lineage.

Interestingly, the phenomenon of post-reproductive lifespan is not confined to human females. It has been observed across a diverse array of female vertebrates, including nonhuman primates [5], certain species of toothed whales [6], and even guppies [7]. For instance, in nonhuman primates, older females often play a crucial role in the social structure of their groups, providing care and support to younger females and their offspring [5,8]. This nurturing behavior likely results in enhanced survival rates for these young ones, thus indirectly benefiting the older females’ genetic legacy [9].

Beyond vertebrates, post-reproductive lifespans have also been documented in the insect world. One notable example is found in the gall-forming aphid *Quadrartus yoshinomiyai*, where post-reproductive females exhibit remarkable self-sacrifice by defending their colony and the reproductive individuals within it [10]. Similarly, in the parthenogenetic ant *Pristomyrmex punctatus*, older worker ants transition from reproductive roles to foraging duties as they age, thus contributing to the colony’s survival [11]. In all the aforementioned cases, with the exception of guppies, the animals in question typically live in kin groups. This social structure likely enhances the indirect fitness benefits they receive from caring for their relatives or their relatives’ offspring [12,13]. By prioritizing the welfare of their kin over their own reproductive opportunities, these individuals may increase the chances of their genetic material being passed on to future generations.

Eusocial insects, including ants and honeybees, further exemplify this concept of extreme altruism driven by kin selection [12,13]. In these societies, individuals often forego their own reproductive opportunities to support the reproductive efforts of their relatives. In a honeybee (*Apis mellifera*) colony, workers are generally sterile when a queen is present, as her pheromones suppress the development of their ovaries [14]. However, in the absence of the queen and her pheromones, honeybee workers can activate their ovaries, allowing them to lay unfertilized eggs that develop into males, known as drones [14,15]. When colonies permanently lose their queen—what is referred to as becoming ‘hopelessly queenless’—the workers’ ability to reproduce becomes critical. Hopelessly queenless colonies typically emerge after the queen dies during her mating flight, an event that occurs with a probability ranging from 14% to 35% [16,17,18]. The drones that are raised in these queenless nests are viable and may have some chance of engaging in mating activities [19,20]. For the worker bees, the period following the queen’s death represents their last opportunity to reproduce before the colony ultimately collapses due to a lack of replacement workers [21,22]. Researchers have proposed that orphaned workers ought to concentrate on their own reproduction and refrain from engaging in challenging and hazardous activities like foraging [23,24]. Nevertheless, earlier studies indicate that honeybee workers without a queen partake in both personal reproduction and various colony duties, such as producing food for the brood, foraging, and defending the colony [25]. Yet, some workers in these queenless colonies remain sterile and do not activate their ovaries, typically attributed to genetic differences among workers from various patrilines [21,26] or to environmental factors that influence worker development [27]. These factors—genetic and environmental—can affect the reproductive capacity of workers by affecting the number of ovarioles in their ovaries, which is closely associated with ovary activation and individual reproductive capability [28]. Another possible explanation for the existence of workers with inactive ovaries in queenless colonies is related to their physiological age at the time the colony loses its queen. Workers that are physiologically older may be less likely to activate their ovaries, as this process is metabolically demanding [29] and there is no assurance of surviving long enough to lay their own eggs. Recent studies have indicated that this explanation may hold some truth. Physiologically older individuals—whether due to advanced age or a shortened lifespan from puncturing—are less likely to activate their ovaries and engage in egg-laying compared to those orphaned at a young age [30]. The authors of this recent paper propose that this phenomenon could be likened to a form of menopause. They suggest that older individuals, rather than investing energy in ovarian activation and their own reproductive capabilities, may instead channel their energy into supporting the fitness of their relatives [30]. However, in the aforementioned study, older individuals never activated their ovaries and instead immediately redirected their resources to assist their relatives. This observation does not entirely align with the traditional definition of menopause, which refers to the cessation of reproductive capacity in female animals, characterized by the end of their fertile cycle and their ability to bear offspring or lay eggs [1,31].

To explore whether menopause is possible in honeybee workers, it would be necessary to investigate how the development of their ovaries changes with age. The study presented below attempted to assess how ovarian activation varies with the physiological age of workers. For this purpose, newly emerged individuals were divided into two groups: one served as a control group, while the other had their lifespan artificially shortened through thorax puncturing, which simulated physiological aging [30,32]. The workers were then marked and released into their native colonies, and the degree of ovarian development was assessed at various intervals—specifically when the workers were 3, 6, 9, 12, 15, 18, 21, 24, 27, and 30 days old. Unfortunately, it is not possible to estimate the degree of ovarian development in the same individuals of different ages, as the preparation is performed on deceased specimens. However, it can be assessed in workers from a single cohort that have lived in an orphaned colony since the start of their adult life. This approach provides insights into the patterns of ovarian activation changes within the entire group.

## 2. Methods

This research was conducted in June and July 2022 at an experimental apiary in Krakow, southern Poland. Three queenright honeybee colonies *(A. m.* carnica) were studied, each consisting of 20,000 to 40,000 workers. All the colonies were treated identically, following the experimental designs of previous studies. Initially, frames with newly emerged bees were transferred from each colony to an incubator set at 36 °C. All workers that emerged within 24 h were divided into two groups: (1) an untreated control group and (2) a group that was injured by puncturing the last segment of the thorax with a needle (diameter: 0.35 mm; puncture depth: until the first drop of hemolymph) to shorten their life expectancy [30,32,33].

Next, 50 workers from each colony and experimental group were placed in wooden frame cages (13 × 9 cm and 5 cm high, with glass and steel mesh sides) and provided with a small piece of bee comb. Two cages were prepared for each experimental colony: one for the control group and one for the injured bees. The cages were incubated at 36 °C and 50–60% relative humidity (RH), and the bees had access to a 50% sucrose solution and water ad libitum. The cages were checked daily, and dead bees were counted and removed. The second portion of the newly emerged workers (0 days old) from both groups was euthanized by freezing for future dissection under a stereomicroscope. The remaining newly emerged workers, both control and injured, were marked on the thorax with a spot of paint that differed in color between the groups (using Marabu Brilliant Painter) and were returned to their colony as soon as possible.

When the marked workers returned to the hive, each colony was temporarily orphaned by removing the queen. Three days after the colonies were orphaned, when the marked workers were 3 days old, 15 workers from both groups (control and injured) were recaptured (the exact number of workers is detailed in the Appendix A). This procedure was repeated when the workers were 6, 9, 12, 15, 18, 21, 24, 27, and 30 days old. The recaptured bees were euthanized by freezing for future dissection under a stereomicroscope.

Workers from all the groups, colonies, and age categories were examined under a stereomicroscope (binocular loupe), and the number of ovarioles, extent of ovary development, and size of the hypopharyngeal gland (HPG) were examined. The size of the HPG was determined by calculating the average size of 10 acini, which are sac-like structures making up the entire gland. This was done by taking the square root of the longest acinus diameter and multiplying it by the shortest diameters of five acini on the right side and five on the left side (refer to Figure 1A) [30]. The total number of ovarioles in both ovaries was recorded for each worker, and the overall ovarian development was evaluated. To characterize the stage of ovary development, the most developed ovariole from each ovary was selected, and its maximum diameter (the widest point) was measured following the method described by Nakaoka [34] (see Figure 1B,C [30]). Additionally, ovary development was classified on the following relative scale, as previously described [35,36]: 1, non-activated ovary; 2, previtellogenic activated ovary; 3, vitellogenic ovary with developing oocytes; and 4, mature ovary with at least one egg. This classification helped analyze the distribution of ovary development across different colonies, age groups, and treatment conditions. The detailed results and related figures are available in the Appendix A file.

A mixed model three-way ANOVA was used to compare parameters (ovariole number, ovariole size, and hypopharyngeal gland size) with respect to worker type (control or injured) and age of workers as fixed effects, while colony was treated as a random effect. If the effect of an experimental treatment was statistically significant, the ANOVA was followed by multiple comparisons using the post hoc Tukey HSD test, with *p* = 0.05 considered significant. Differences in survival between control and injured workers in cages and hives were also analyzed using the generalized linear model (GLZ) module with a Poisson distribution and log link function. Here, colony was treated as a random effect, and worker type was a fixed effect. If a factor was statistically significant, the GLZ was followed by multiple comparisons using the post hoc Tukey HSD test, with *p* = 0.05 indicating significance. All the calculations were performed using Statistica 13.3.

A mixed-model three-way ANOVA was used to compare parameters (ovariole number, ovariole size, and hypopharyngeal gland size) with respect to worker type (control or injured) and the age of workers as fixed effects, while colony was treated as a random effect. If the effect of an experimental treatment was statistically significant, the ANOVA was followed by multiple comparisons using the post hoc Tukey HSD test, with *p* = 0.05 considered significant. Differences in survival between control and injured workers in cages and hives were also analyzed using the generalized linear model (GLZ) module with a Poisson distribution and log link function. Here, colony was treated as a random effect, and worker type was a fixed effect. If a factor was statistically significant, the GLZ was followed by multiple comparisons using the post hoc Tukey HSD test, with *p* = 0.05 indicating significance. All the calculations were performed using Statistica 13.3.

## 3. Results

The lifespan of bees in the cages did not differ between workers coming from different colonies (Wald’s χ^2^ = 3.10, *p* = 0.212; Figure 2), but it depended on the experimental group (control or injury; Wald’s χ^2^ = 21,240.32, *p* < 0.001; Figure 2) Similar to previous studies [30,32,33], we found that the control workers had a longer lifespan than those that were injured. The mean longevity ranged from 13.0 to 13.7 days for control workers and from 9.7 to 10.6 days for injured individuals, depending on the experimental colony (Figure 2).

The HPG of workers depends on both the experimental treatment (control or injury) and the age of the workers (see Table 1). Gland size was not affected by the colony of origin. Moreover, the size of the HPG gland varied with age, and it differed between control and injured workers. The size of the HPG after emergence and up to the sixth day of life for the workers was similar in both the control and injured individuals, and it increased in both groups (Figure 3). However, in the injured workers, there was a drastic decrease in the size of the HPG, while in the control group, the size of the gland continued to increase until the 15th day of life, stabilized, and only began to decrease around the 21st day. The size of the gland in both groups equalized by the time the workers reached the 27th day of life.

The number of ovarioles did not differ among workers from different colonies or treatments (control and injured) and was not influenced by the age of the workers (see Table 2). However, the size of the ovaries was dependent on the colony from which the workers originated and their age, while the experimental treatment (control or injured workers) had no significant effect on the size of the ovaries (see Table 3 and Figure 4). In both punctured and control workers, the size of the ovaries reached a maximum between 15 and 21 days of life and then decreased to a level similar to that observed immediately after emergence (Figure 4).

## 4. Discussion

Consistent with our predictions, injured honeybee workers in the cage experiment exhibited a significantly shorter lifespan (3 days shorter) compared to untreated workers from the control groups (Figure 2 and Appendix A). This finding aligns with existing research that indicates injuries can adversely affect insect longevity [30,32,33]. The underlying reasons likely include an increased susceptibility to infections, which can be exacerbated by physical damage [37,38]. Injuries may compromise the bees’ immune systems, making it more difficult for them to fend off pathogens [39]. Additionally, the metabolic costs associated with healing and recovery could further deplete their energy reserves [40], ultimately reducing their overall lifespan.

After confirming that workers from various age and treatment groups had different lifespans, we moved on to explore whether these groups also varied in anatomical characteristics. Our initial focus was on the hypopharyngeal glands (HPGs), which are essential for producing and storing brood food [41,42]. Generally, young honeybees have large HPGs characterized by high protein synthesis rates, whereas older foragers possess smaller, less active glands [41]. In line with previous research [30,32], our findings indicated that the HPGs were significantly larger in physiologically younger workers, while older bees exhibited a notable decrease in gland size. Furthermore, we observed significant differences between injured and control workers. For the injured bees, the size of the HPG increased only until the sixth day of life, after which it began to decline rapidly. In contrast, control workers demonstrated a consistent increase in gland size until the fifteenth day of life, with a decrease starting only on the twenty-first day (Table 1; Figure 3; Appendix A). These results are consistent with established studies on honeybee colonies, where workers typically start foraging between 18 and 28 days into adulthood [14]. However, workers with shorter lifespans, such as our injured bees, tend to begin foraging at an earlier age compared to control workers with typical life expectancies. This earlier onset of foraging activities can significantly affect the development of their HPGs [14,32,41,42].

The next step in our research was to examine how the size of the ovaries changes with the age of worker honeybees. We began by investigating whether there were any differences in the number of ovarioles among individuals from different families, ages, and experimental treatments (injured and control groups). The number of ovarioles in a honeybee’s ovaries is established during the larval stage [14] and is influenced by various factors, including the quality and quantity of food provided [14], the presence of a queen [36], and the influence of her mandibular gland pheromones during development [43]. Typically, the number of ovarioles remains stable in adult honeybee workers, with declines occurring only in rare instances [44]. However, ovary activation, which is crucial for reproduction, occurs during the adult life of the workers and is influenced by a combination of factors [45]. These include the presence of a queen [46] and her pheromones [47], the existence of brood, the tasks performed by the workers [35,48], quality of food [49], and, importantly, the number of ovarioles in the ovaries themselves [28]. Previous studies have demonstrated a positive correlation between the number of ovarioles and ovary activation in honeybee workers, suggesting that a higher number of ovarioles may facilitate reproductive processes [28]. Given this background, we aimed to determine whether our experimental workers displayed any variations in ovariole number. Our results indicated that there were no significant differences in this parameter across all the tested factors in our experiment. This finding suggests that the number of ovarioles does not vary significantly between the different groups, which led us to conclude that this particular parameter did not influence the subsequent and more critical result regarding the size of the ovaries in the tested bees.

The most intriguing finding of this study is the observation that older individuals deactivate their ovaries. In both the control and injured groups, the size of the ovaries in worker bees increases to a certain point before beginning to decrease, ultimately reaching sizes similar to those of newly emerged bees. This result can be interpreted as a form of menopause in bees, indicating a cessation of reproductive activity in older individuals. However, it was found that the injury had no statistically significant effect on the activation of the ovaries in the workers. The graph, along with the data showing the averages in the Appendix A, reveals that the injured workers reached maximum ovary development by the 15th day of life, whereas the control individuals achieved this by the 18th day. Despite these observations, there were no statistical differences in ovary development between the bees from these two groups. This raises the question of why the puncturing, which reduced the lifespan of the workers by three days, did not significantly impact the differences in ovarian activation. This can be explained quite simply: the activation and deactivation of the ovaries occur gradually and exhibit considerable variability [14,34,35,45]. At any given age, there can be individuals with fully activated ovaries (with eggs present in the ovarioles) alongside those that are in the process of activation but have not yet begun producing eggs. As a result, the variance in measurements is quite large, and even individuals from adjacent age groups often do not show significant differences in ovary size. Therefore, the three-day difference in lifespan may not have had a meaningful impact on ovarian activation, especially since, in the absence of a queen in the nest, there remains an impetus to activate the ovaries anyway.

The phenomenon of post-reproductive lifespan in insects extends beyond honeybees and has been documented in various species, including the clonal gall-forming aphid, *Q. yoshinomiyai* [10]. In this species, post-reproductive females exhibit a self-sacrificial behavior to defend the colony and its reproductive individuals. This behavior is particularly notable because, in aphid colonies, all members are genetically identical clones, which eliminates the conflicts of kin selection that typically arise in sexually reproducing species. This unique genetic structure has likely facilitated the evolution of post-reproductive altruism among these aphids [10]. Another compelling example can be found in the parthenogenetic ant, *P. punctatus*. In this species, as colony members age, they tend to shift their roles from reproduction to foraging and assisting their sisters in reproductive efforts [11]. This transition underscores the adaptability and social complexity of insect societies, highlighting how older individuals can contribute to the colony’s overall success in ways other than direct reproduction

Our findings resonate with the grandmother hypothesis, which seeks to explain the evolutionary origins of post-reproductive lifespans in various animal species [1,4]. This hypothesis posits that within kin groups, the presence of post-reproductive individuals—analogous to grandmothers—can boost the fitness of the group by assisting their reproductive relatives, originally adult daughters in the case of humans. In the context of honeybees, workers who may not have enough time to activate their ovaries still play a vital role in their colony’s success by aiding fellow colony members, both full sisters and half-sisters. Consequently, this shift in focus from personal reproduction to the welfare of the colony exemplifies the intricate social dynamics and evolutionary strategies that characterize honeybee communities, reflecting a broader trend observed in various insect societies.

## 5. Conclusions

The findings highlight that honeybee workers experience a reproductive transition with age, resembling menopause in other species. As they prioritize colony welfare, older bees may forgo personal reproduction, underscoring the complexity of social dynamics and kin selection within honeybee communities.

## Figures and Tables

**Figure 1 animals-15-01402-f001:**
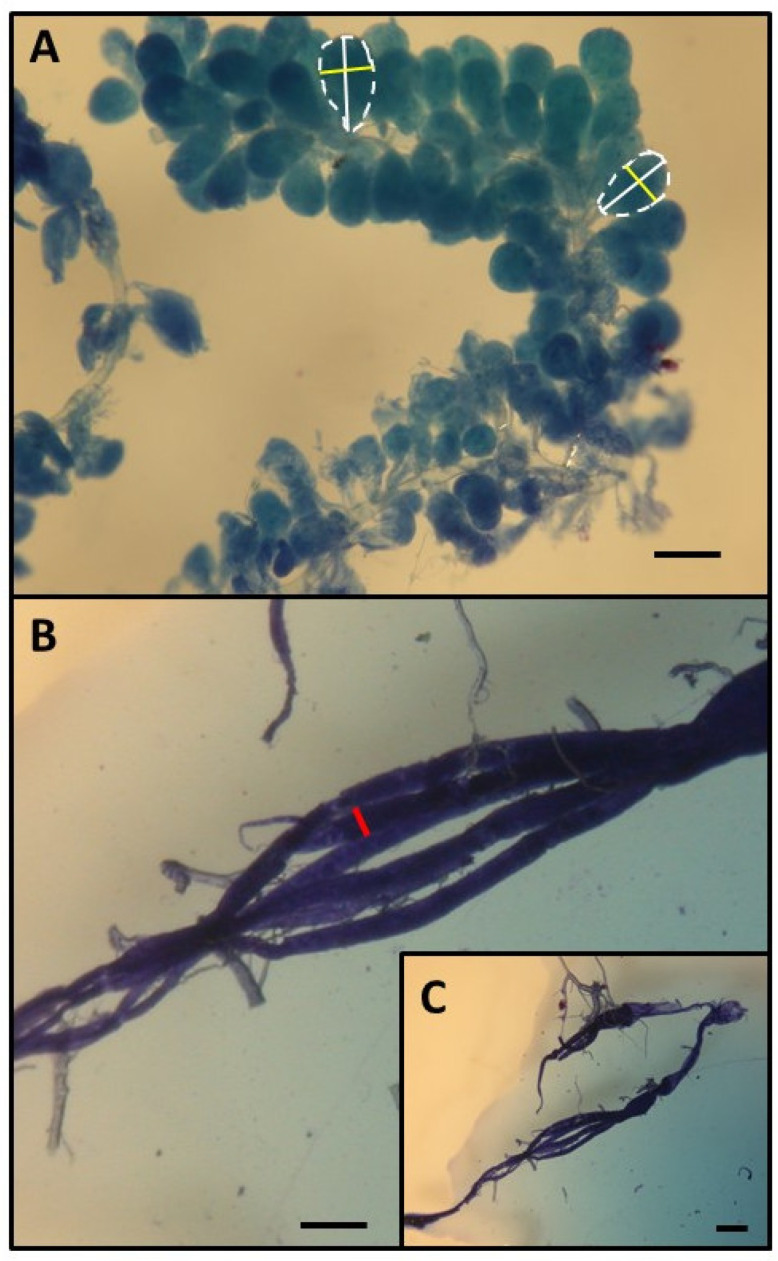
Assessment of HPG (**A**) and ovary size (**B**,**C**) (see also [30]). The size of the hypopharyngeal gland (HPG) was determined by averaging the sizes of 10 acini, calculated as the square root of the longest acinus diameter multiplied by the shortest diameters of five acini from the right gland and five from the left gland [30]. Each individual acinus is outlined with a dashed line, with the longer diameters marked by a white bar and the shorter diameters by a yellow bar. To evaluate ovarian development, the most developed ovariole from each ovary was selected, and the maximum diameter (shown by a red bar) of the two ovarioles was measured as their greatest width. Scale bars represent 100 µm.

**Figure 2 animals-15-01402-f002:**
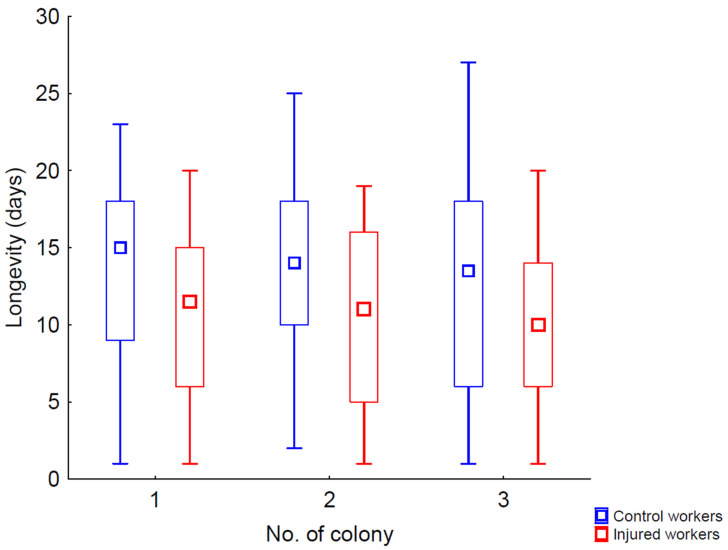
Longevity (median, quartiles, and range) of control (blue bars) and injured (red bars) workers in the cage.

**Figure 3 animals-15-01402-f003:**
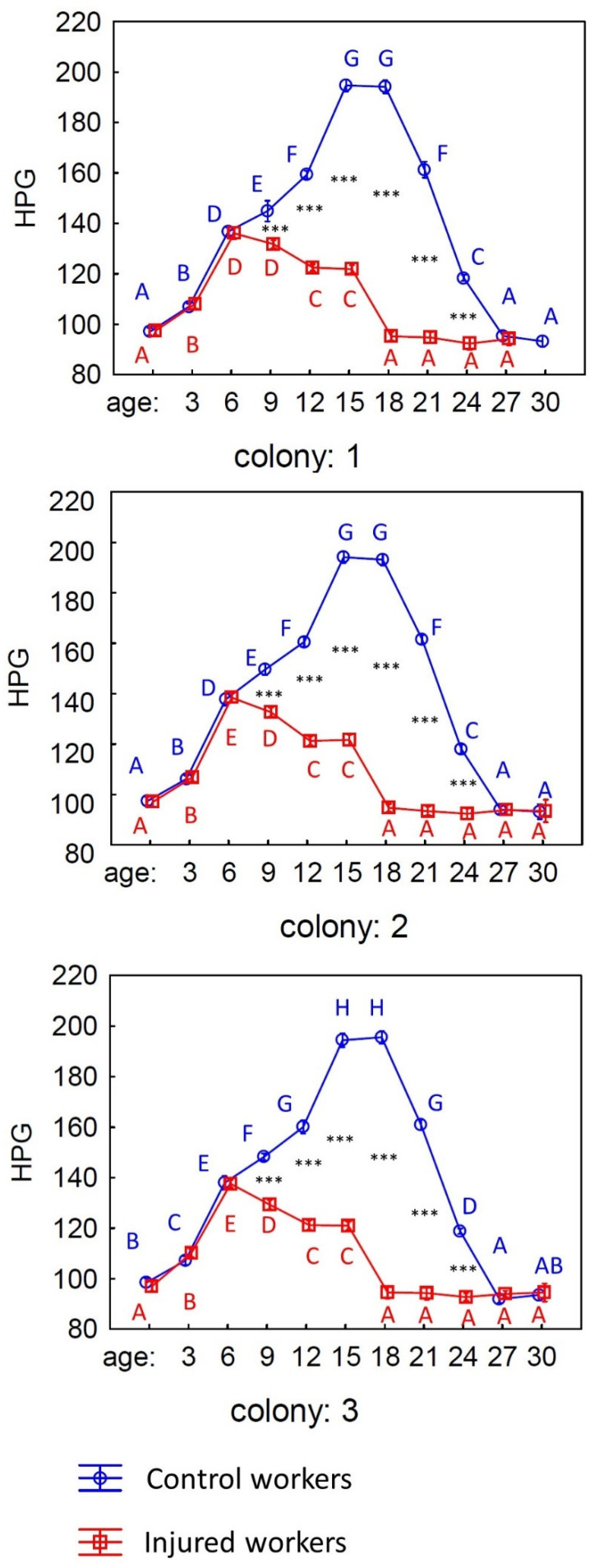
The size of the hypopharyngeal gland (HPG) of workers of different ages from three experimental colonies: control (blue line) and injured workers (red line). The different blue letters indicate significant differences in HPG size among workers of various ages in the control group, while the red letters show the differences in HPG size among workers of different ages in the injured group. The black stars indicate differences between workers from the control and injured groups of the same age.

**Figure 4 animals-15-01402-f004:**
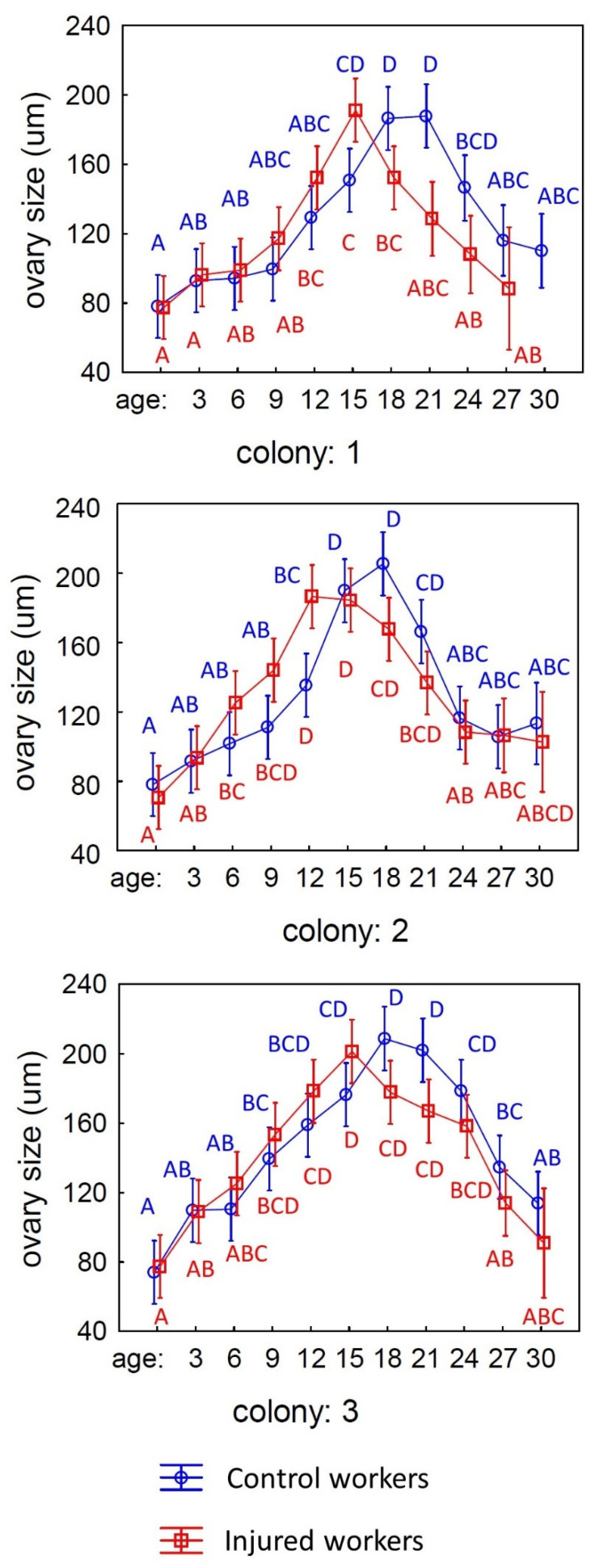
The size of the ovary development of workers of different ages from three experimental colonies: control (blue line) and injured workers (red line). The different blue letters indicate significant differences in ovary size among workers of various ages in the control group, while the red letters show the differences in ovary size among workers of different ages in the injured group.

**Table 1 animals-15-01402-t001:** The results of the three-way ANOVA for the size of the HPG.

	Effect (F/R)	SS	Degr. of Freedom	MS	Den.Syn. Error df	Den.Syn. Error MS	F	*p*
Intercept	Fixed	12,197,200	1	12,197,200	5.9568	1.11953	10,894,963	0.000000
Colony	Random	1	2	1	2.2938	10.67207	0	0.940422
Age	Fixed	405,213	10	40,521	20.3319	15.97009	2537	0.000000
Treatment	Fixed	172,080	1	172,080	2.3328	7.96339	21,609	0.000011
Colony * age	Random	320	20	16	19.3393	12.89786	1	0.319215
Colony * treatment	Random	15	2	8	20.6224	12.88767	1	0.557920
Age * treatment	Fixed	266,187	10	26,619	20.1321	12.89145	2065	0.000000
Colony *age * treatment	Random	245	19	13	852.0000	12.58334	1	0.428249
Error		10,721	852	13				

**Table 2 animals-15-01402-t002:** The results of the three-way ANOVA for the number of ovarioles.

	Effect (F/R)	SS	Degr. of Freedom	MS	Den.Syn. Error df	Den.Syn. Error MS	F	*p*
Intercept	Fixed	15242.16	1	15242.16	2.1431	0.961790	15847.70	0.000035
Colony	Random	1.93	2	0.97	0.0000			
Age	Fixed	3.34	10	0.33	22.3264	0.245854	1.36	0.261033
Treatment	Fixed	0.73	1	0.73	3.9368	0.144161	5.05	0.088933
Colony * age	Random	4.70	20	0.24	19.7836	0.371684	0.63	0.842550
Colony * treatment	Random	0.22	2	0.11	22.8221	0.386314	0.28	0.759051
Age * treatment	Fixed	3.13	10	0.31	21.6474	0.380891	0.82	0.612494
Colony * age * treatment	Random	6.98	19	0.37	852.0000	0.823350	0.45	0.980575
Error		701.49	852	0.82				

**Table 3 animals-15-01402-t003:** The results of the three-way ANOVA for the size of ovarioles.

	Effect (F/R)	SS	Degr. of Freedom	MS	Den.Syn. Error df	Den.Syn. Error MS	F	*p*
Intercept	Fixed	14,152,702	1	14,152,702	2.0059	33000.62	428.8617	0.002291
Colony	Random	68,482	2	34,241	5.8845	3270.95	10.4682	0.011528
Age	Fixed	1,089,478	10	108,948	20.2087	2873.23	37.9182	0.000000
Treatment	Fixed	3178	1	3178	2.1594	1637.41	1.9411	0.289633
Colony * age	Random	57,784	20	2889	19.3800	1187.67	2.4327	0.028045
Colony * treatment	Random	3307	2	1654	20.8205	1191.17	1.3882	0.271678
Age * treatment	Fixed	123,825	10	12,382	20.2694	1189.87	10.4066	0.000006
Colony * age * treatment	Random	22,547	19	1187	852.0000	1295.69	0.9159	0.562966
Error		1,103,930	852	1296				

## Data Availability

The datasets generated during and/or analyzed during the current study are available in the Appendix A and also from the corresponding author upon reasonable request.

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
