# Peer review of "Post-Reproductive Lifespan in Honey Bee Workers with Varying Life Expectancies"

_animals, 2025, doi:10.3390/ani15101402_

Round 1
Reviewer 1 Report
Comments and Suggestions for Authors
The research paper “Post-Reproductive Lifespan in Honey Bee Workers with Varying Life Expectancies” aims to examines the post-reproductive lifespan and ovarian activation in honeybee workers (Apis mellifera) with differing life expectancies. It was well organized and presented. However, there are some concerns that I suggested the author to address to make it more clear.
Line 9, 18 and 34, “Apis mellifera” should be italicized.
Line 128: “A. m. carnica” should be italicized.
In the Introduction, “Unfortunately, it is not possible to assess the degree of ovarian activation in surviving bees” (line 121), could you briefly explain why here?
Line 143-146: how long did the workers marking last? Were the workers from each colony emerged within 24 hours enough for the subsequent sample colletion?
Line 147-150: the 3 days old workers were kept in queen-less colonies, how about the others workers? After the marked workers returned to the hive, the colonies remained queen-less until all the workers at different age were collected?
In the Methods, you mentioned that you assessed the ovary development on a relative scale. But in the Results, there was no information about this.
Line 252-253: “After confirming that workers from various age and treatment groups had different lifespans”, I understand that you confirmed the workers from different treatment groups (control and injured) had different lifespans, but how about the various age group?
Author Response
Reviewer #1
The research paper “Post-Reproductive Lifespan in Honey Bee Workers with Varying Life Expectancies” aims to examines the post-reproductive lifespan and ovarian activation in honeybee workers (Apis mellifera) with differing life expectancies. It was well organized and presented. However, there are some concerns that I suggested the author to address to make it more clear.
Line 9, 18 and 34, “Apis mellifera” should be italicized.
Thank you for this suggestion — it has been addressed in the new version of the MS.
Line 128: “A. m. carnica” should be italicized.
Thank you for this suggestion — it has been addressed in the new version of the MS.
In the Introduction, “Unfortunately, it is not possible to assess the degree of ovarian activation in surviving bees” (line 121), could you briefly explain why here?
Thank you for pointing this out. The original sentence was imprecise. What I meant was that it is impossible to assess ovarian development in the same individuals at different ages, since capturing and dissecting them results in their death. Therefore, individuals from the same cohort but at different ages were selected to estimate variability in ovarian development. To clarify, the sentence has been revised to:
“Unfortunately, it is not possible to estimate the degree of ovarian development in the same individuals of different ages, as the preparation is performed on deceased specimens. However, it can be assessed in workers from a single cohort that has lived in an orphaned colony since the start of their adult life.” (line 121-124 of new version of MS)
Line 143-146: how long did the workers marking last? Were the workers from each colony emerged within 24 hours enough for the subsequent sample colletion?
Marking bees with a marker is a common practice in beekeeping. In apiaries, queen bees are often marked in this way (they live up to three years, and the marking remains visible); the marker is not toxic to them. Regarding the number of bees, during the May-June period, a queen lays about 2,000 eggs per day, usually on one or two combs in close proximity. This allowed for the collection of large sample sizes. I initially estimated that I would have about 30 individuals per group and age (with 11 age groups, resulting in a total of 660 individuals), which could be easily obtained. However, since the bees lived naturally within their colonies and were exposed to various threats, including natural mortality, more individuals were marked: colony 1 - 1,556 bees; colony 2 - 1,643 bees; colony 3 - 1,378 bees. (WINSTON, Mark L. The biology of the honey bee. harvard university press, 1991)
Line 147-150: the 3 days old workers were kept in queen-less colonies, how about the others workers? After the marked workers returned to the hive, the colonies remained queen-less until all the workers at different age were collected?
The text states that all colonies were orphaned (line ……of new version of MS), and then marked bees were released into them. The next step was to catch bees living in the orphaned colonies on the 3rd, 6th, 9th, 12th, 15th, 18th, 21st, 24th, 27th, and 30th day of their life to observe how ovarian development changes over time. The colonies were orphaned because, only in such colonies, do workers have the opportunity to activate their ovaries; in colonies with a queen, the queen inhibits ovarian activation in workers.( WINSTON, Mark L. The biology of the honey bee. harvard university press, 1991)
In the Methods, you mentioned that you assessed the ovary development on a relative scale. But in the Results, there was no information about this.
Some studies have used only a relative scale, but this has changed over time, and now the degree of ovarian activation is more commonly assessed by measuring the width of the widest ovariole. However, to allow comparison with earlier studies, I performed such an assessment and included it in the supplement as original data—should anyone wish to make comparisons.
Line 252-253: “After confirming that workers from various age and treatment groups had different lifespans”, I understand that you confirmed the workers from different treatment groups (control and injured) had different lifespans, but how about the various age group?
Older individuals generally have a shorter expected lifespan (when considered across the entire cohort, not just a single individual) compared to younger ones. Of course, in the context of research, everything depends on how long the individuals naturally live, because I agree that if an animal lives for 3 years, a difference of 3 days may be negligible. For honeybee workers, this is not a significant issue during the season, because workers in May-June (the period of the experiment) live just over a month .( WINSTON, Mark L. The biology of the honey bee. harvard university press, 1991). Therefore, a 3-day difference accounts for about 7-8% of their lifespan. During the course of this experiment, there was already a high risk that I would not have 30-day-old individuals, as they might not survive that long.
Reviewer 2 Report
Comments and Suggestions for Authors
Paper submitted by Kuszewska presents original results on the reproductive status and longevity of honeybee workers after injury. It adds valuable information on the dynamics of reproduction in hives, especially after the death of the queen.
As I can judge, experimental protocols were well conducted, and statistitic treatments are apprioriate (traditionnal Anovae and Khi square, GLM for survival). I was surprised that body size was not considered as a fixed effect. Was it recorded?
Text is clear, well writen and organized.
In discussion, some considerations could improve the impact of the ms:
- add (if available) some comparisons with bumble bees that are known to produce microcolonies from isolated workers
- Precise what could be the role of the so called 'menopaused workers' in the success of an orphan colony. Especially, considerations are missing on the metabolic activity of those 'retreated' workers. In other words, we can understand that they are almost dead.
Minor points:
- keywords must differ from the title, delete 'honeybee', 'post-reproductive lifespan' and add 'grand mother hypothesis' and 'ovaries' (or other keywords)
- In not sure 'personal reproduction' is clear, change to 'individual reproduction'
- Was is possible to assess the ovary stage in dead workers? Set it in Materials and Methods
- Line 190, change 'from differ colony' to 'from different colonies'
- Line 265 needs a reference for foraging age
- lines 267-268, Could the reverse be also considered: the development of HPGs could affect foraging?
- lines 289-290, is the number of ovaries fixed during pupal development? If yes it may not vary after complete development, and does not depends on the worker life history (see https://doi.org/10.1534/genetics.109.105452)
Author Response
Reviewer #2
Paper submitted by Kuszewska presents original results on the reproductive status and longevity of honeybee workers after injury. It adds valuable information on the dynamics of reproduction in hives, especially after the death of the queen.
As I can judge, experimental protocols were well conducted, and statistitic treatments are apprioriate (traditionnal Anovae and Khi square, GLM for survival). I was surprised that body size was not considered as a fixed effect. Was it recorded?
Body mass was not monitored; however, the bees were randomly assigned to experimental groups (the first 5 as controls, the next 5 were punctured, and the procedure was repeated until each group of 50 individuals was completed in the cages). This means that even if body mass influences lifespan, it likely increased the variability, making it more difficult to detect differences between the control and punctured individuals.
Text is clear, well writen and organized.
In discussion, some considerations could improve the impact of the ms:
add (if available) some comparisons with bumble bees that are known to produce microcolonies from isolated workers
Unfortunately, I do not have such data, but definitely, such a comparison would be interesting.
Precise what could be the role of the so called 'menopaused workers' in the success of an orphan colony. Especially, considerations are missing on the metabolic activity of those 'retreated' workers. In other words, we can understand that they are almost dead.
According to derived genetic studies, which do not activate their ovaries (initially for no apparent reason) in other functions, they may care for their sisters' offspring or induce food sharing within the colony Kuszewska, K.; Woloszczuk, A.; Woyciechowski, M. Reproductive Cessation and Post-Reproductive Lifespan in Honeybee Workers. Biology 2024, 13, 287, doi:10.3390/biology13050287, as previously mentioned in lines 331-337 of the manuscript.
Minor points:
keywords must differ from the title, delete 'honeybee', 'post-reproductive lifespan' and add 'grand mother hypothesis' and 'ovaries' (or other keywords)
Thank you. I have updated this in the new version of my manuscript: I deleted "honeybee" and added "grandmother hypothesis" and "ovaries."
In not sure 'personal reproduction' is clear, change to 'individual reproduction
Thank you, I chaned it in new version of MS
Line 190, change 'from differ colony' to 'from different colonies'
Thank you, I chaned it in new version of MS
Line 265 needs a reference for foraging age
Thanks, I added this
lines 267-268, Could the reverse be also considered: the development of HPGs could affect foraging?
The size of the hypopharyngeal gland is related to the performance of worker bees as nurse bees. When worker bees begin foraging, the glands regress. Of course, younger bees, or more precisely those with a longer life expectancy—even as foragers—have better developed hypopharyngeal glands than older individuals with a shorter life expectancy, because in the latter, the degeneration process has been ongoing for a longer period (Kuszewska, K., & Woyciechowski, M. (2013). Reversion in honeybee, Apis mellifera, workers with different life expectancies. Animal Behaviour, 85(1), 247-253.; Amdam, G. V., Aase, A. L. T., Seehuus, S. C., Fondrk, M. K., Norberg, K., & Hartfelder, K. (2005). Social reversal of immunosenescence in honey bee workers. Experimental gerontology, 40(12), 939-947.; Robinson, G. E., Page Jr, R. E., Strambi, C., & Strambi, A. (1992). Colony integration in honey bees: mechanisms of behavioral reversion. Ethology, 90(4), 336-348.). It is these younger individuals that can revert and reactivate their glands. While it cannot be entirely ruled out that the development of the HPG influences reversion, it is known that if individuals are required to perform nurse duties for a longer period (such as during a sudden increase in larvae), the gland is more developed in older bees WINSTON, Mark L. The biology of the honey bee. harvard university press, 1991.; Robinson, G. E., Page Jr, R. E., Strambi, C., & Strambi, A. (1992). Colony integration in honey bees: mechanisms of behavioral reversion. Ethology, 90(4), 336-348. Therefore, it is more plausible to suggest that another factor, which correlates with both the tendency to revert and the degree of HPG development, is involved.
lines 289-290, is the number of ovaries fixed during pupal development? If yes it may not vary after complete development, and does not depends on the worker life history (see https://doi.org/10.1534/genetics.109.105452)
Yes, the number of ovarioles is determined during the larval stage and does not change throughout the individual's life. However, the total number of ovarioles is correlated with life-history parameters such as ovarian activation (individuals with a greater number of ovarioles are more frequently active; Makert, G. R., Paxton, R. J., & Hartfelder, K. (2006). Ovariole number—a predictor of differential reproductive success among worker subfamilies in queenless honeybee (Apis mellifera L.) colonies. Behavioral ecology and sociobiology, 60, 815-825.) and, in certain cases, with lifespan(Kuszewska, K., Miler, K., Rojek, W., & Woyciechowski, M. (2017). Honeybee workers with higher reproductive potential live longer lives. Experimental Gerontology, 98, 8-12.). Therefore, we wanted to ensure that individuals from all experimental groups (treatment and age) have, on average, the same number of ovarioles to rule out their influence on the results.